# Position: Explainable AI is Causal Discovery in Disguise

## Abstract

Explainable AI (XAI) has intrigued researchers since the earliest days of artificial intelligence. However, with the surge in AI-based applications—especially deep neural network models—the complexity and opacity of AI models have intensified, renewing the call for explainability. As a result, an overwhelming number of methods have been introduced, reaching a point where surveys now summarize other surveys on XAI. Yet, significant challenges persist, including unresolved debates on accuracy-explainability tradeoffs, conflicting evaluation metrics, and repeated failures in sanity checks. Further complications arise from fairness violations, robustness issues, privacy concerns, and susceptibility to manipulation. While there's broad agreement on the importance of XAI, expert panels and major conferences continue to reveal that *the only consensus on how to achieve it is a lack of one*. This has led some to question whether the discord stems from a fundamental absence of ground truth for defining *"the"* correct explanation.

This position paper argues that explainable AI is, in fact, a supervised problem—albeit with a target rooted in a profound, often elusive, understanding of reality – in this sense, XAI is causal discovery in disguise. By reframing XAI queries as causal inquiries—whether about *data*, *models*, or *decisions*—we prove the necessity and sufficiency of causal models for XAI, encouraging community convergence around advanced methods for concept and causal discovery, potentially through interactive, approximate causal inference. We contend that without such a model, XAI remains limited by its lack of ground truth, keeping us entrenched in uncertainty.

[1]Anonymous Institution, Anonymous City, Anonymous Region, Anonymous Country. Correspondence to: Anonymous Author <anon.email@domain.com>.

Preliminary work. Under review by the International Conference on Machine Learning (ICML). Do not distribute.

## 1. Introduction

As early as the 1980s, the challenge of explainable AI (XAI) has been recognized as both critical and ambiguously defined (Kodratoff, 1994). Numerous attempts to tackle this issue have led to a diverse array of methods, which are organized and categorized across various surveys. Notable works include those focusing specifically on neural networks (Yosinski et al., 2015; Montavon et al., 2018; Samek et al., 2021), as well as broader surveys addressing explainable AI in general (Doshi-Velez & Kim, 2017; Došilović et al., 2018; Hoffman et al., 2018; Guidotti et al., 2018; Lipton, 2018; Adadi & Berrada, 2018; Gilpin et al., 2018; Miller, 2019; Gunning et al., 2019; Du et al., 2019; Tjoa & Guan, 2020; Arrieta et al., 2020; Carvalho et al., 2019; Murdoch et al., 2019). With each of these survey papers exceeding 1,000 citations, it's perhaps enough to warrant a survey of surveys (Speith, 2022).

Despite the very many attempts, the field continues to grapple with fundamental questions. The definitions of *explainability* and *interpretability* may not always be agreed upon (Preece et al., 2018; Ehsan & Riedl, 2024; Leblanc & Germain, 2024; Namatevs et al., 2022; Marcinkevičs & Vogt, 2020), and debates over accuracy-explainability tradeoffs have split the community into proponents of *inherent* vs. *post-hoc* explainability approaches (Rudin, 2019; Gunning & Aha, 2019; Laugel et al., 2019). The lack of consensus over definitions and methodologies is further compounded by concerns over fairness (Von Kügelgen et al., 2022), robustness (Yeh et al., 2019; Ghorbani et al., 2019; Kindermans et al., 2019; Hamon et al., 2020), privacy violations (Shavit & Moses, 2019b), and the susceptibility of explanations to being manipulated or fooled (Dombrowski et al., 2019; Shavit & Moses, 2019a; Heo et al., 2019; Slack et al., 2020; Sullivan & Verreault-Julien, 2022; Wickstrøm et al.). Due to the lack of ground truth explanations, the community has been compelled to pursue an axiomatic framework for defining explainability (Sundararajan et al., 2017; Janizek et al., 2021; Amgoud & Ben-Naim, 2022), yet, despite their axiomatic appeal, later work has shown failures in essential sanity checks (Adebayo et al., 2018; Tomsett et al., 2020; Karimi et al., 2023).

**This position paper posits that the persistent discord in XAI arises not from an absent ground truth but from a**

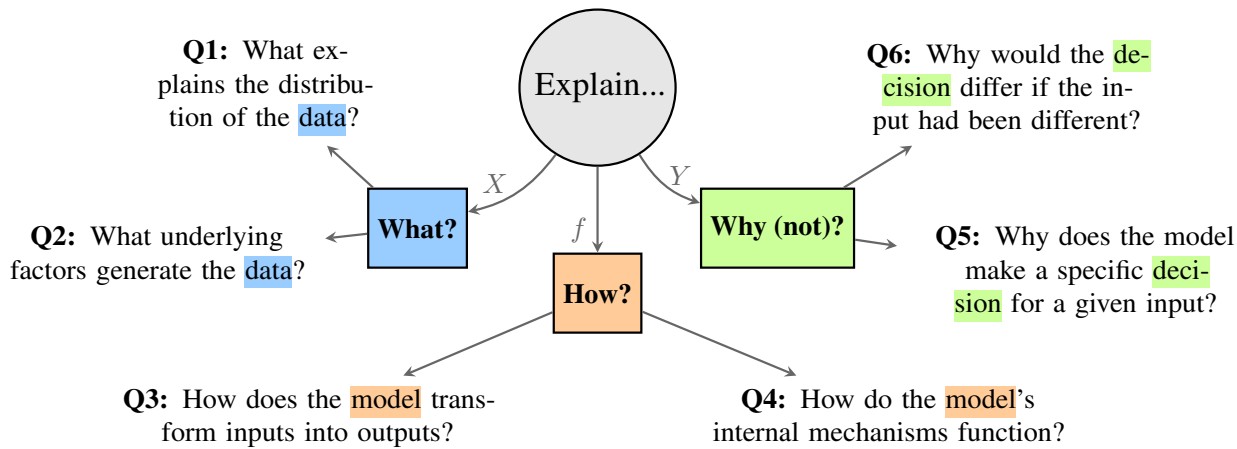

Figure 1: Core methods in XAI for explaining $f : X \rightarrow Y$ are categorized by purpose into data-based ("What?", $X$), model-based ("How?", $f$), and decision-based ("Why (not)?", $Y$) questions.

**ground truth that exists, albeit as an elusive and challenging target: the *causal model* that governs the world.** While acknowledging the difficulty of obtaining this world model, we argue that the real barrier to consensus in XAI lies in the field's near-total disregard for actively seeking it. Causal assumptions, we contend, are essential to bring coherence to XAI by addressing core questions through a principled lens; without such assumptions, XAI methods risk providing explanations that lack rigor, reliability, or generalizability. Motivations for interpretability are diverse: some practitioners use XAI to debug data or models, others need it for regulatory compliance or trust-building with end-users, and yet others seek actionable insights for interventions. The *purpose* for seeking interpretability therefore shapes the specific explanatory methods chosen. For instance, debugging data often requires unveiling biased patterns, whereas actionable insights require identifying causal drivers of outcomes.

Several studies have highlighted the importance of causality in XAI, identifying specific areas where a causal foundation could improve existing methods. Karimi et al. (2020; 2021) advocate for incorporating causal relationships into counterfactual explanations to enable actionable outcomes, while Chou et al. (2021) and Baron (2023) critique existing counterfactual methods for lacking causal grounding, which they argue leads to spurious correlations and incomplete explanations. Similarly, Carloni et al. (2023) highlight the absence of causality in current XAI as a critical limitation, emphasizing its necessity for building trust in AI systems. Finally, Beckers (2022) highlights causality's potential for action-guiding explanations in XAI, and Chen et al. (2023) propose integrating causal discovery into XAI methods to enhance interpretability, leading to more actionable explanations. Our aim is to unify and expand on these insights to emphasize that almost all XAI approaches, from model attributions to concept-based methods, implicitly demand causal reasoning.

In the following sections, we categorize existing XAI methods based on the purpose of questions they address, illustrating specific areas where causal assumptions clarify and enhance each approach. We then review various causal frameworks for formally grounding these methods, demonstrating that causal assumptions are not only *sufficient* but *necessary* for rigorous, reliable, and generalizable explanations in XAI. We also discuss how causal representation learning tasks underpin these approaches, bridging recent work in circuit-based interpretability and abstraction to show the breadth of causal discovery's impact on XAI. We conclude with future research suggestions.

## 2. Background on Explainable AI (XAI)

To understand the role of causality in explainable AI (XAI), we first categorize existing XAI methods based on the primary purpose of their explanations: the data $X$, the model $f$,[1] or the decisions $Y$. As in Figure 1, these questions can be organized into three categories of questions based on purpose:

- **Data-based ("What?")**: Uncovering the structure and significance of the data $X$.
- **Model-based ("How?")**: Exploring how the model $f$ transforms input $X$ into output $Y$.
- **Decision[2]-based ("Why (not)?")**: Interpreting specific output $Y$ for given input $X$ and model $f$.

By structuring XAI methods within this framework, we highlight gaps due to a lack of causal grounding, setting a foundation for our argument that causality is essential for rigorous, valid XAI.

---

[1]Here, $f$ represents the predictive model to be explained, distinct from the causal model of the world, $\mathcal{M}$.

[2]Like Miller (2019), we use "decision" to refer broadly to AI system outputs, such as categorizations or action choices.

## 2.1. Data-Based Interpretability ("What?")

Data-based interpretability focuses on understanding the structure and characteristics of the input data $X$, answering questions such as:

**Q1**: "What explains the distribution of the data?"

**Q2**: "What underlying factors generate the data?"

Data-based interpretability methods are particularly useful for exploratory data analysis and in contexts where understanding biases or clusters within the data is crucial. Example methods include:

- **Attention Mechanisms** (Vaswani et al., 2017) are widely used in neural networks, especially transformers, allowing the model to dynamically focus on relevant parts of the input $X$ for each prediction. This highlights which components of $X$ the model finds important, providing insights into the data structure and dependencies therein.
- **Dimensionality Reduction** (e.g., PCA, t-SNE (Van der Maaten & Hinton, 2008)) maps high-dimensional data $X$ to a lower-dimensional space, revealing structures and clusters. This helps identify key patterns and assess their impact on predictions.

These approaches align closely with the principles of *causal discovery*, which aims to identify the causal relationships and dependencies within the data itself (Spirtes et al., 2001; Pearl, 2009). By revealing the structure of $X$ and uncovering influential features, these methods help illuminate underlying patterns that may influence model behavior. For example, clustering and dimensionality reduction techniques highlight significant groupings and trends within the data, while attention mechanisms focus on key features that contribute to predictions. Such methods provide an essential foundation in XAI, as understanding the causal dependencies within $X$ aids in detecting data biases and ensuring robust performance in the model's outputs.

## 2.2. Model-Based Interpretability ("How?")

Model-based interpretability seeks to explain the function $f$, specifically how the model processes input $X$ to produce output $Y$. This category addresses questions such as:

**Q3**: "How does the model transform inputs into outputs?"

**Q4**: "How do the model's internal mechanisms function?"

Model-based interpretability is essential in regulatory and high-stakes environments where transparency into $f$'s workings is required. These methods include:

- **Feature Interaction Methods** (e.g., Partial Dependence Plots (Friedman, 2001), Accumulated Local Effects (Apley & Zhu, 2020)) explore interactions within $f$ by showing how different features affect $Y$. Partial Dependence Plots, for instance, illustrate the effect of one or two features on $Y$ while other features are kept constant, revealing interactions in $f$.
- **Feature Attribution Methods** (e.g., LIME (Ribeiro et al., 2016), SHAP (Lundberg & Lee, 2017)) decompose $f(X)$ to assign an importance score to each feature in $X$, indicating its contribution to the output $Y$. Some works interpret these attributions as estimates of local (individual) causal effects (Chattopadhyay et al., 2019), suggesting that LIME can be approximated via input gradients in sufficiently smooth regions.
- **Saliency and Visualization Methods** (e.g., Saliency Maps (Simonyan et al., 2013), Grad-CAM (Selvaraju et al., 2016)) visualize gradients to identify important regions in $X$ that affect $Y$, such as which image pixels are influential in a prediction. Grad-CAM, for example, generates a heatmap highlighting image regions that impact the model's output.
- **Surrogate and Simplified Models** aim to approximate complex models $f$ in specific regions using *inherently interpretable* models (e.g., decision trees, linear models). Towell & Shavlik (1993) extract rules to enhance interpretability in neural networks, and LIME provides local explanations through linear models (Ribeiro et al., 2016). While MASALA adapts locality for improved fidelity (Anwar et al., 2024), MaLESCaMo introduces causal surrogate models (Termine et al., 2023), and Laugel et al. (2018) focus on locality for surrogates in post-hoc interpretability.
- **Model-Intrinsic Interpretability Approaches** use interpretable models like linear models, decision trees, and rule-based systems, allowing direct inspection of $f$'s parameters to understand how $X$ maps to $Y$ without post-hoc explanations. For instance, Generalized Additive Models (GAMs) model responses as sums of functions of predictors (Hastie & Tibshirani, 1987). The Bayesian Case Model uses representative cases for interpretability (Kim et al., 2014), while the Bayesian Rule Set framework learns interpretable rule sets (Wang et al., 2017). Interpretable Decision Sets provide a joint framework for description and prediction, facilitating comprehensible decision-making processes (Lakkaraju et al., 2016).

These approaches are closely related to understanding *causal mechanisms* (Pearl, 2000; Peters et al., 2017)—the specific processes through which changes in input features $X$ influence the output $Y$. By attributing importance to features, analyzing interactions, and approximating internal model logic, these methods help uncover the pathways within $f$ that drive model predictions. For example, feature attribution methods quantify each feature's contribution to $Y$, aligning with causal mechanisms by revealing how

particular inputs influence the model's output. Similarly, saliency maps and feature interaction methods highlight key regions and feature dependencies within $f$, providing an interpretative view of how the model operates. This mechanistic understanding is essential in domains where transparent explanations are required, as it allows stakeholders to see not only which features matter but also how they interact to produce predictions.

### 2.3. Decision-Based Interpretability ("Why (not)?")

Decision-based interpretability focuses on explaining specific outputs $Y$ for given inputs $X$ and model $f$, addressing questions such as:

**Q5**: "Why does the model make a specific decision for a given input?"

**Q6**: "Why would the decision differ if the input had been different?"

Decision-based interpretability is valuable in applications where understanding the rationale behind individual decisions and possible alternatives is crucial, such as in personalized recommendations or legal judgments. Example methods include:

- **Counterfactual and Example-based Methods** (Wachter et al., 2017) illustrate what minimal changes to $X$ would be necessary to alter the output $Y$, providing insight into decision boundaries by showing hypothetical scenarios in which the decision would differ.

- **Post-hoc Concept-based Explanation Methods** (e.g., TCAV) (Kim et al., 2018) explain $Y$ in terms of high-level human-defined concepts, rather than individual features of $X$. TCAV, for example, assesses the relevance of specific concepts (like "striped" or "curved") to a prediction, offering an interpretable, concept-level explanation.

These methods draw on concepts from *actual causality* (Halpern, 2016) by using *counterfactual reasoning* to explore why a particular outcome was reached. Halpern and Pearl's causal model formalizes this approach, defining causes through counterfactual dependencies that clarify necessary and sufficient conditions for an outcome (Halpern & Pearl, 2005). In practical terms, answering "why" questions involves identifying the minimal changes in $X$ that would alter $Y$, thereby uncovering the causal factors influencing the decision. Counterfactual reasoning provides actionable insights, as it clarifies the conditions under which an alternative outcome could occur. This concept of causality has also been extended by Woodward (2005), who argues that interventions and counterfactuals provide a foundation for understanding causal explanations and model behavior. By leveraging such causal insights, decision-based interpretability approaches not only highlight decision bound-

aries but also enhance understanding of model outcomes and potential user actions. This purpose-driven categorization of data-based, model-based, and decision-based XAI methods structures the response to XAI questions posed in Figure 1. However, lacking causal assumptions limits robustness and generalizability across contexts. Below, we introduce causal foundations and explore how causal models address these XAI gaps. We will also see how existing lines of circuit-based interpretability and causal abstraction further strengthen the claim that *explanation is causal discovery in disguise*.

## 3. Background on Causality

Causality aims to model the relationships between variables where one variable causes changes in another, thereby going beyond mere statistical correlations to capture the underlying mechanisms of the data-generating process. Unlike correlations, causal relationships entail directional influence, allowing one to predict the effect of interventions and counterfactuals in the system (Pearl, 2009). Multiple frameworks formalize causality, including the Potential Outcomes framework (Rubin, 2005), Graphical Models (Spirtes et al., 2001), and Structural Causal Models (SCMs) (Pearl, 2009), each offering unique perspectives on understanding causation. For the purposes of this work, we adopt Pearl's SCM framework, as it provides a rigorous formalism for reasoning about causal mechanisms, interventions, and counterfactuals—critical components for constructing XAI systems. We formalize the claim that access to the true causal model, represented as an SCM, is both *sufficient* and *necessary* for addressing purpose-driven methods on the "What?", "How?", and "Why (not)?" of explanations.

### 3.1. Preliminaries

To ground our claims, we define key concepts and notations employed throughout this section.

**Definition 3.1 (Structural Causal Model (SCM))** *An SCM $\mathcal{M}$ is a tuple $\langle \mathbf{U}, \mathbf{V}, \mathbf{F}, P(\mathbf{U}) \rangle$, where:*

- $\mathbf{U} = \{U_1, U_2, \ldots, U_m\}$ *is a set of exogenous (unobserved) variables.*
- $\mathbf{V} = \{V_1, V_2, \ldots, V_n\}$ *is a set of endogenous (observed) variables.*
- $\mathbf{F}$ *is a set of structural equations $f_V : V \in \mathbf{V}$, where each $f_V$ maps the parents of $V$ and relevant exogenous variables to $V$, i.e., $V = f_V(\mathrm{pa}(V), U_V)$. $\mathbf{F}$ specifies the causal mechanisms underlying the data-generating process, providing a mechanistic description of causal relationships.*
- $P(\mathbf{U})$ *is a joint probability distribution over the exogenous variables $\mathbf{U}$.*

**Definition 3.2 (Causal Graph)** *The causal graph $\mathcal{G}$ associated with an SCM $\mathcal{M}$ is a directed acyclic graph (DAG) where nodes represent variables in $\mathbf{V}$, and edges represent direct causal relationships as specified by the structural equations in $\mathbf{F}$. This graph provides a visual representation of causal dependencies and is a fundamental tool for identifying causal pathways and potential confounders (Spirtes et al., 2001; Pearl, 2009).*

**Definition 3.3 (Observ., Interv., and Counterf. Queries)** *Access to an SCM $\mathcal{M}$ enables analysis involving three primary types of queries, each offering unique insights into the relationships captured by $\mathcal{M}$:*

- *Observational Queries: These involve probabilities computed from the observed data distribution $P(\mathbf{V})$. They describe associations between variables as observed without external manipulation and are limited to capturing correlations rather than causation.*

- *Interventional Queries: Interventions modify the underlying structural equations in $\mathbf{F}$ to estimate causal effects. Such interventions are denoted by the do-operator, $do(\cdot)$, representing an exogenous alteration that severs the usual dependence of a variable on its causal parents, allowing for predictions under manipulated conditions. For example, the query $P(Y = y | do(X = x))$ estimates the probability of $Y = y$ when $X$ is set to $x$ by intervention (Pearl, 2009).*

- *Counterfactual Queries: Counterfactual queries explore hypothetical scenarios that diverge from observed reality, posing "what if" questions about alternative outcomes. For a given observed outcome, counterfactual reasoning considers what the outcome would have been had certain variables taken different values. This requires conditioning on observed data to infer observed values exogenous variables, $U = u$, and then modifying variables, $X = x'$, to then predict $Y_{X=x'}(u)$ counterfactuals (Rubin, 2005; Pearl, 2009).*

**Definition 3.4 (Causal Discovery)** *Unlike the queries above which presuppose a causal model, causal discovery (Spirtes & Zhang, 2016; Malinsky & Danks, 2018; Glymour et al., 2019; Nogueira et al., 2022; Eberhardt, 2017; Vowels et al., 2022) aims to infer the causal graph $\mathcal{G}$ from observational or experimental data, an essential step for constructing accurate causal models. This process faces challenges, including latent confounders, data scarcity, and reliance on assumptions like causal sufficiency. Methods for causal discovery include constraint-based approaches (e.g., PC algorithm) (Spirtes et al., 2001), score-based methods (Huang et al., 2018), and functional causal models (e.g., additive noise models) (Peters et al., 2017). The ability to uncover causal relationships is crucial for XAI, as it directly affects the fidelity of the explanations generated.*

## 4. Sufficiency and Necessity of Causality for Explainable AI

In the following theorems, we first formalize the sufficiency claim, followed by the necessity claim.

**Definition 4.1 (Accurate and Complete Answers to Q1-6)** *Following Pearl (2009), we say an answer to any of the six core XAI questions (Q1–Q6 in Figure 1) is* accurate *and* complete *if it coincides exactly with what the* **true** *Structural Causal Model (SCM) $\mathcal{M}$ predicts for that query. Concretely:*

- *Observational correctness (Q1, Q2): The distribution of observed variables and the underlying generating factors match those in $\mathcal{M}$.*
- *Interventional correctness (Q3, Q4): The effect of manipulating inputs or tracing internal mechanisms reflects the causal structure of $\mathcal{M}$.*
- *Counterfactual correctness (Q5, Q6): The counterfactual outcome $Y_{X=x'}(u)$ for a specific exogenous state $u$ matches the counterfactual computed under $\mathcal{M}$.*

**Theorem 4.2 (Sufficiency of the True SCM for XAI)** *Let $\mathcal{M} = \langle \mathbf{U}, \mathbf{V}, \mathbf{F}, P(\mathbf{U}) \rangle$ be the unique true Structural Causal Model of the data-generating process. Under standard assumptions (acyclicity, no unmeasured confounders, well-defined exogenous variables), having full access to $\mathcal{M}$ is* **sufficient** *to provide accurate and complete answers to the six core XAI questions (Q1–Q6) depicted in Figure 1.*

**Proof Sketch** *(Full proof in App. A)* Since $\mathcal{M}$ specifies:

1. The causal graph $\mathcal{G}$ over the endogenous variables $\mathbf{V}$,
2. A set of structural equations $\mathbf{F}$ indicating how each $V_i \in \mathbf{V}$ depends on its parents $\mathrm{pa}(V_i)$ and possibly exogenous $U_{V_i}$,
3. The distribution $P(\mathbf{U})$ over the exogenous variables,

it uniquely determines the joint distribution of all variables, any interventional distribution via the do-operator $do(\cdot)$, and any counterfactual query via abduction–action–prediction (Pearl, 2009). Mapping these distributions to to Q1–Q6:

- **Q1 (Distribution of data) and Q2 (Underlying factors).** The law of structural models allows us to derive $P(\mathbf{V})$ exactly from $\mathcal{M}$, and we see how exogenous variables $\mathbf{U}$ and functions $f_{V_i}$ generate the observed data.
- **Q3 (How does the model process inputs?) and Q4 (How do internal mechanisms operate?).** By tracing causal pathways in $\mathcal{G}$ (and applying $\mathbf{F}$ iteratively), we reveal how input $X$ propagates to output $Y$ through intermediate variables (hidden layers or sub-modules).
- **Q5 (Why a specific decision?) and Q6 (Why would the decision differ?).** Given $(X = x, Y = y)$, we infer

exogenous $\mathbf{u}$ (abduction), modify $X \leftarrow x'$ (action), and compute $Y_{X=x'}(\mathbf{u})$ (prediction). This explains both why the model made its decision and how it would change under a different input.

Because $\mathcal{M}$ yields precise observational, interventional, and counterfactual results, it provides complete and accurate explanations for all six questions. Thus, knowing the true SCM is *sufficient* for XAI. ∎

### Theorem 4.3 (Necessity of the True SCM for XAI)

*Suppose a dataset $\mathbf{V}$ is generated by a* true *but unknown SCM $\mathcal{M}$. If an alternative model $\hat{\mathcal{M}}$ does not match $\mathcal{M}$ in at least one structural equation or in its exogenous distribution $P(\mathbf{U})$, then there exists at least one of the six XAI questions (Q1–Q6) for which $\hat{\mathcal{M}}$ cannot provide an accurate and complete answer.*

**Proof Sketch** *(Full proof in App. A)* Recall that accurate and complete answers require reproducing *exactly* the observational, interventional, or counterfactual results from $\mathcal{M}$. We prove by contradiction:

1. **Assume** $\hat{\mathcal{M}}$ is a different SCM than $\mathcal{M}$ but still claims to yield correct answers for *all* Q1–Q6.

2. There are three broad query types:
   - **Observational (Q1–Q2)**: If $\hat{\mathcal{M}}$ differs in $\mathbf{F}$ or $P(\mathbf{U})$, it may induce a different joint distribution over $\mathbf{V}$, contradicting Q1 or misidentifying underlying data factors (Q2).
   - **Interventional (Q3–Q4)**: Even if $\hat{\mathcal{M}}$ matches observationally, the do-operator $\mathrm{do}(X=x)$ can produce different outcomes in $\hat{\mathcal{M}}$ vs. $\mathcal{M}$ due to differences in causal structure or confounding assumptions (Pearl, 2009).
   - **Counterfactual (Q5–Q6)**: Counterfactual questions rely on abduction–action–prediction with the *true* exogenous state. A mismatch in structural equations leads to different counterfactual results.

3. Hence, there must be at least one question Q1–Q6 where $\hat{\mathcal{M}}$'s answer diverges from $\mathcal{M}$'s. This contradicts the assumption that $\hat{\mathcal{M}}$ is correct for *all* XAI questions.

Therefore, to guarantee accuracy and completeness across all six questions simultaneously, access to the true SCM $\mathcal{M}$ is *necessary* for XAI. ∎

### 4.1. Discussion on Robustness and Limitations

Theorems 4.2 and 4.3 assume access to the *true* Structural Causal Model (SCM) $\mathcal{M} = \langle \mathbf{U}, \mathbf{V}, \mathbf{F}, P(\mathbf{U}) \rangle$, ensuring accurate and complete answers to XAI questions (Q1–Q6). However, in real-world applications, such oracle-level causal knowledge is rarely accessible. Instead, we must rely on estimated models $\hat{\mathcal{M}}$ that approximate $\mathcal{M}$,

introducing challenges in accuracy and reliability.

**Partial Causal Knowledge and Sensitivity Analysis.** Since complete causal knowledge is typically unavailable, practitioners often incorporate known causal relationships into $\hat{\mathcal{M}}$ to refine explanations beyond purely statistical methods. This approach reduces reliance on spurious correlations but does not guarantee correctness. To assess robustness, sensitivity analysis (Saltelli et al., 2004) can quantify the stability of explanations under small perturbations to $\hat{\mathcal{M}}$. However, even systematic robustness checks cannot ensure validity when the underlying model is fundamentally misspecified.

**Challenges in Learning the Causal Model (Causal Discovery).** An alternative approach is to infer $\mathcal{M}$ via causal discovery methods, but this presents several key challenges:

- **Faithfulness and Causal Sufficiency.** Causal discovery typically assumes *faithfulness* (i.e., observed independencies reflect true causal structure) and *causal sufficiency* (i.e., no hidden common causes). If these assumptions fail, the inferred causal structure may be incorrect.[3]

- **Sample Complexity and Computational Constraints.** Even when causal sufficiency holds, reliable causal discovery requires a large sample size, especially in high-dimensional settings. The number of samples required grows exponentially with the number of variables, making exhaustive search computationally infeasible (Kalisch & Bühlman, 2007).

- **Identifiability and Equivalence Classes.** Even with unlimited data and valid assumptions, causal discovery methods often recover only a Markov equivalence class of DAGs—multiple causal graphs that imply the same observational dependencies (Spirtes et al., 2001). This ambiguity means that without interventional data, key causal relationships may remain unresolved.

**Takeaway.** The ideal of fully accurate and complete XAI, as established in Theorems 4.2–4.3, is difficult to achieve due to the limitations above. In light of these challenges, correlation-based explanations (e.g., feature importances, saliency maps) may suffice when the goal is to detect patterns, biases, or anomalies rather than to enable interventions. Nonetheless, a more nuanced view is that the *required level of causal grounding* depends on the stakeholder's objective. When reliability matters—such as in high-stakes decision-making—approximate causal models, even if imperfect, can yield explanations that address the diverse interpretability questions in Fig. 1.

---

[3]Consider three variables $(X, Y, Z)$ where $Z$ is an unmeasured confounder influencing both $X$ and $Y$ (i.e., $Z \rightarrow X$, $Z \rightarrow Y$). Without observing $Z$, the learned $\hat{\mathcal{M}}$ may wrongly suggest a direct causal link between $X$ and $Y$, leading to incorrect explanations.

# 5. A Way Forward

Recognizing these challenges, we propose strategic directions to address them, focusing on two interrelated tasks: *Concept Discovery* and *Relation Discovery*. By advancing methods in these areas, we can approximate causal models more effectively and enhance explainable AI. Despite the limitations, we encourage the community to embrace these challenges, as they are essential steps toward realizing that *explainable AI is, in essence, causal discovery in disguise*.

## 5.1. Dual Challenges in Causal XAI: Concept Discovery and Relation Discovery

**Concept Discovery.** Effective explanations require a shared language of interpretable concepts $\{Z_i\}$ that align with the stakeholder's understanding. Explanations should be constructed using well-defined, semantically clear variables to ensure meaningful communication. Current XAI methods vary along a *Concept-Alignment Spectrum*:

- **Fully Specified Concepts:** At one end, methods like SHAP (Lundberg & Lee, 2017) and causal recourse (Karimi et al., 2021) provide explanations using features $X_i$ with direct semantic meaning, such as age or income. These methods produce mappings $\phi : X \to \mathbb{R}$ that quantify feature contributions and support actionable interventions.
- **Low-Level Features:** At the other end, methods like saliency maps (Simonyan et al., 2013) highlight groups of pixels in images, which lack inherent semantic meaning and require abstraction to align with human concepts.
- **Concept-Based Methods:** In the middle, methods like TCAV (Kim et al., 2018) attempt to align explanations with predefined concepts by measuring alignment with existing embeddings. However, TCAV is limited to known concepts and cannot discover new, relevant concepts—the "unknown unknowns"—that may be crucial for understanding the model's behavior.

To enhance concept discovery, we advocate for methods that can uncover and represent new concepts, potentially via causal approaches such as Concept Bottleneck Models (Koh et al., 2020), Causal Concept Effect (Goyal et al., 2019), and Neuro-Symbolic Concept Learners (Mao et al., 2019; Ellis et al., 2023) which offer promising directions by treating concepts as entities that facilitate action and interpretability. These methods enable both structured learning and deeper understanding by integrating causal reasoning into concept discovery. Moreover, concepts should be identified at a granularity that is *useful* to a given stakeholder (or audience). Even if we had a perfect SCM of low-level features (e.g., pixels), explanations would remain unhelpful unless translated to higher-level abstractions that align with human mental models (Rubenstein et al., 2017; Beckers & Halpern,

2019). Future research should thus emphasize learning and *serving* these causal concepts at the right level of detail, possibly via user interaction or iterative refinement.

**Relation Discovery.** Discovering causal relationships among identified variables $\{V_i\}$ remains a foundational challenge in interpretability. Traditional causal discovery and structure learning aim to infer a directed acyclic graph $\mathcal{G} = (\mathbf{V}, \mathbf{E})$, where $\mathbf{V}$ represents variables and $\mathbf{E}$ captures causal dependencies. Established algorithms like PC (Spirtes et al., 2001) and score-based methods (Huang et al., 2018) provide structure but are often computationally demanding in high-dimensional settings. We propose leveraging advances in *causal representation learning* (Bengio et al., 2019; Schölkopf et al., 2021), which strive to capture both the concept space and causal structure. These approaches can deepen interpretability by jointly learning representations that are both semantically meaningful and causally informative. However, the scalability challenge becomes especially pressing for large-scale or high-dimensional models (e.g., LLMs). Here, purely symbolic or conditional-independence-based causal discovery can be prohibitively slow. Exploring approximations such as sparse regressions, online structure learning, or domain-guided heuristics (Granger, 1969) may be necessary to handle real-world data at scale.

## 5.2. Leveraging Approximate Models and Interactive Approaches

In practice, obtaining a fully accurate causal model is often infeasible due to data and computational limitations. To address this, we advocate for *approximate causal models* supplemented by interactive, user-driven methods. By iteratively refining causal structures through user feedback and interventions, approximate models can better align with real-world needs, enabling users to validate and adjust causal assumptions as needed.

In scenarios where full causal structure discovery is impractical, *interactive approaches* enable iterative refinement of causal models based on user interactions and counterfactual queries. This user-in-the-loop methodology aligns with recent advances in chain-of-thought reasoning (Wei et al., 2022) and large language models (e.g., GPT-4), allowing explanations to evolve with stakeholder feedback, enhancing their relevance and causal grounding. Moreover, an interactive process can reveal the "right" level of abstraction for each user's goals (Teso et al., 2023), acknowledging that an exhaustive model of the world is neither feasible nor desirable for most tasks. Instead, explanations should focus on *those causal factors* that the user can understand and act upon, effectively capturing a subset of the world's SCM aligned with the user's mental model (Gerstenberg et al., 2021; Gerstenberg, 2024).

### 5.3. Summary of Recommendations

Despite its limitations, our core thesis remains: *explainable AI is causal discovery in disguise*. Advancing methods in concept and relation discovery will enable the construction of approximate causal models that enhance the rigor, reliability, and applicability of explanations. We encourage the community to invest in:

1. **Developing Robust Causal Discovery Algorithms:** Improving methods to better handle high-dimensional data, hidden confounders, and model misspecification. Future work should also explore multi-level abstractions (Rubenstein et al., 2017; Beckers & Halpern, 2019) to balance expressivity with user interpretability.

2. **Advancing Causal Representation Learning:** Jointly learning concepts and causal relations that are interpretable, stable, and scalable. Concept discovery should align with users' internal models, recognizing that no single "correct" variable decomposition exists universally (Teso et al., 2023). This calls for methods that bridge machine-learned representations with human-understandable structures.

3. **Promoting Interactive Explanations:** Engaging stakeholders in refining causal models through iterative feedback. This aligns with "explanatory interactive learning" (Teso & Kersting, 2019), where users refine models by correcting explanations, steering causal learning to ensure relevance and actionability.

By pursuing these directions, we can mitigate the practical challenges of causal XAI while moving toward a principled foundation for explainability.

### 5.4. Alternative Views: The Limitations of SCMs for Representing Human Intuition

While we argue that explainable AI is fundamentally a problem of causal discovery, an opposing perspective questions whether Structural Causal Models (SCMs) are the right framework for capturing human reasoning and intuition. Specifically, SCMs are often criticized for their limited expressiveness in representing rich, structured mental models of the world. Human reasoning frequently operates through *intuitive theories* (Gerstenberg & Tenenbaum, 2017), which go beyond the propositional nature of SCMs. For example, in physics, people intuitively understand the world in terms of objects, forces, and attributes (e.g., mass, elasticity, friction), rather than abstract causal graphs. When reasoning counterfactually, humans naturally ask questions such as "What if this object hadn't been there?" or "What if a reasonable person had acted differently?"—queries that are difficult to formalize in an SCM, where variables typically represent discrete events or predefined states. Unlike SCMs, which encode causal mechanisms as structured equations

over variables, human cognition often blends causal reasoning with spatial, temporal, and qualitative constraints, making it unclear whether SCMs are the best mathematical framework for modeling how people construct and interpret explanations.

One response to this challenge is to extend SCMs with hierarchical abstractions that align with how humans structure knowledge. Recent work on *causal abstraction models* and *neuro-symbolic reasoning* offers promising directions by introducing layers of representation that move beyond traditional SCM constraints. However, these approaches remain an open area of research, and critics argue that a truly human-aligned XAI framework may require fundamentally different tools—potentially drawing from cognitive science, probabilistic programs, or physics-inspired models—to bridge the gap between mechanistic causality and intuitive human understanding.

### 5.5. Conclusion

The vast landscape of explainable AI is marked by an overwhelming number of methods, surveys, and perspectives, all of which underscore the field's current lack of consensus. This paper argues that achieving such consensus hinges on reframing XAI as causal discovery, demonstrating through formal necessity and sufficiency results that *causal assumptions* are both essential and adequate to address purpose-driven questions around the "What?", "How?", and "Why (not)?" of explanations. By positioning explanations within a causal model, researchers and practitioners can align on clearer, more robust foundations for XAI, effectively viewing it as *causal discovery in disguise*.

Building on this viewpoint, we advocate for advancing *concept discovery* and *relation discovery* to identify variables and causal links at a level of abstraction that matches stakeholders' mental models. In practice, approximate causal modeling and interactive refinement are key. By iteratively engaging users (e.g., through counterfactual queries or explanatory interactive learning), we can converge on a causal representation that offers actionable insights while accommodating the complexities of real-world systems.

Ultimately, we encourage the community to see beyond fragmented XAI methods and move toward a unified causal framework—one that embraces multi-level abstractions, interactive approaches, and real-world constraints. Although challenges like scalability, incomplete domain knowledge, and unmeasured confounders remain, they should be viewed not as barriers but as opportunities to refine and extend causal discovery methodologies for explainable AI. By doing so, we believe the field can progress toward a shared, actionable approach to XAI that balances rigor, utility, and adaptability for diverse stakeholders.

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

## A. Sufficiency and Necessity of Causality for Explainable AI

**Theorem 4.2 (Sufficiency of the True SCM for XAI)**
*Let $\mathcal{M} = \langle \mathbf{U}, \mathbf{V}, \mathbf{F}, P(\mathbf{U}) \rangle$ be the unique true Structural Causal Model of the data-generating process. Under standard assumptions (acyclicity, no unmeasured confounders, well-defined exogenous variables), having full access to $\mathcal{M}$ is **sufficient** to provide accurate and complete answers to the six core XAI questions (Q1–Q6) depicted in Figure 1.*

**Proof** We proceed by examining each question individually, using the predefined variables and formal language.

> **Q1: What explains the distribution of the data?**
>
> The SCM $\mathcal{M}$ specifies the structural equations $\mathbf{F}$ and the distribution $P(\mathbf{U})$. The joint distribution of the endogenous variables $\mathbf{V}$ can be derived from $\mathcal{M}$ using the *law of structural models*:
>
> $$P(\mathbf{V}) = \int_{\mathbf{U}} \prod_{V_i \in \mathbf{V}} \delta\left(V_i - f_{V_i}(\mathrm{pa}(V_i), U_{V_i})\right) P(\mathbf{U})\, d\mathbf{U}$$
>
> where $\delta(\cdot)$ is the Dirac delta function ensuring that $V_i$ satisfies its structural equation, and $\mathrm{pa}(V_i)$ are the parents of $V_i$ in the causal graph $\mathcal{G}$ associated with $\mathcal{M}$. Since we can derive $P(\mathbf{V})$ from $\mathcal{M}$, we can fully explain the distribution of the data, accounting for all dependencies and relationships specified by the structural equations and exogenous distributions.

> **Q2: What underlying factors generate the data?**
>
> In the SCM $\mathcal{M}$, the exogenous variables $\mathbf{U}$ represent the underlying factors that are not determined within the model but affect the endogenous variables through the structural equations. Each endogenous variable $V_i$ is generated by: $V_i = f_{V_i}(\mathrm{pa}(V_i), U_{V_i})$. Access to $\mathcal{M}$ gives both the exogenous variables $\mathbf{U}$ and the structural equations $\mathbf{F}$, allowing us to identify and understand the underlying factors generating the observed data.

> **Q3: How does the model transform inputs into outputs?**
>
> Suppose the AI model takes inputs $\mathbf{X} \subseteq \mathbf{V}$ and produces outputs $\mathbf{Y} \subseteq \mathbf{V}$. The causal pathways from $\mathbf{X}$ to $\mathbf{Y}$ are specified in the causal graph $\mathcal{G}$ associated with $\mathcal{M}$. The structural equations define how each variable depends on its parents: $V_i = f_{V_i}(\mathrm{pa}(V_i), U_{V_i})$. By following these equations along the paths from $\mathbf{X}$ to $\mathbf{Y}$, we can trace how inputs are transformed into outputs through the model. Specifically, we can compute the effect of $\mathbf{X}$ on $\mathbf{Y}$ by recursively evaluating the structural equations.

> **Q4: How do the model's internal mechanisms function?**
>
> Internal mechanisms (e.g., hidden layers, intermediate computations) are represented by intermediate endogenous variables $\mathbf{H} \subseteq \mathbf{V}$ in the SCM. The structural equations for the internal variables are: $H_j = f_{H_j}(\mathrm{pa}(H_j), U_{H_j})$ By analyzing these equations and their dependencies, we can understand how the internal variables operate and contribute to the processing of inputs $\mathbf{X}$ to outputs $\mathbf{Y}$. The causal graph $\mathcal{G}$ of model $\mathcal{M}$ shows the connections between $\mathbf{X}$, $\mathbf{H}$, and $\mathbf{Y}$, allowing us to trace the flow of information and causation through the model's internal structure.

> **Q5: Why does the model make a specific decision for a given input?**
>
> Given a specific input $\mathbf{X} = \mathbf{x}$ and the observed output $\mathbf{Y} = \mathbf{y}$, we can perform *abduction* to infer the values of the exogenous variables $\mathbf{U} = \mathbf{u}$ consistent with these observations. Using the inferred $\mathbf{u}$ and the structural equations $\mathbf{F}$, we can then trace the causal pathways from $\mathbf{X} = \mathbf{x}$ to $\mathbf{Y} = \mathbf{y}$, identifying the causal mechanisms and intermediate variables that led to the decision.

> **Q6: Why would the decision differ if the input had been different?**
>
> To answer this counterfactual question, we consider an alternative input $\mathbf{X} = \mathbf{x}'$ while keeping the inferred exogenous variables $\mathbf{U} = \mathbf{u}$ fixed at the values inferred during abduction. Finally, comparing the counterfactual output $\mathbf{Y}^*$ with the original output $\mathbf{Y} = \mathbf{y}$ to understand how and why the decision would differ under the alternative input.

**Overall Conclusion** In each case, access to the true SCM $\mathcal{M}$ provides sufficient information—whether through computing distributions, tracing causal pathways, or performing counterfactual reasoning—to accurately and completely answer each of the six XAI questions.

∎

**Theorem 4.3 (Necessity of the True SCM for XAI)**
*Suppose a dataset $\mathbf{V}$ is generated by a true but unknown SCM $\mathcal{M}$. If an alternative model $\hat{\mathcal{M}}$ does not match $\mathcal{M}$ in at least one structural equation or in its exogenous distribution $P(\mathbf{U})$, then there exists at least one of the six XAI questions (Q1–Q6) for which $\hat{\mathcal{M}}$ cannot provide an accurate and complete answer.*

**Proof** We will demonstrate that without causal information—specifically, without access to the true SCM $\mathcal{M}$—it is impossible to answer the six core XAI questions. We

proceed by addressing each question individually, using the predefined variables and formal language established earlier.

### Q1: What explains the distribution of the data?

Without causal information, we only have access to the observational distribution $P(\mathbf{V})$ of the endogenous variables $\mathbf{V}$. However, $P(\mathbf{V})$ encodes statistical associations but not causal relationships. Statistical dependencies in $P(\mathbf{V})$ can arise from various causal structures, such as direct causation, confounding, or even collider effects.

**Illustrative Example**: Consider three variables $X, Y$, and $Z$ with the following causal structures:

1. **Confounding**: $Z$ is a common cause of $X$ and $Y$, i.e., $Z \to X$, $Z \to Y$.
2. **Causal Chain**: $X$ causes $Z$, which in turn causes $Y$, i.e., $X \to Z \to Y$.
3. **Collider**: $X$ and $Y$ both cause $Z$, i.e., $X \to Z \leftarrow Y$.

All these structures can produce similar statistical associations between $X$ and $Y$ in $P(\mathbf{V})$. Without causal assumptions or knowledge of the underlying SCM, we cannot distinguish among these possibilities.

### Q2: What underlying factors generate the data?

In an SCM $\mathcal{M} = \langle \mathbf{U}, \mathbf{V}, \mathbf{F}, P(\mathbf{U}) \rangle$, the exogenous variables $\mathbf{U}$ and structural equations $\mathbf{F}$ define how the observed data $\mathbf{V}$ are generated: $V_i = f_{V_i}(\mathrm{pa}(V_i), U_{V_i})$, $\forall V_i \in \mathbf{V}$ Without access to $\mathcal{M}$, we lack knowledge of both $\mathbf{U}$ (the unobserved factors) and $\mathbf{F}$ (the causal mechanisms). Consequently, we cannot accurately model the data-generating process.

### Q3: How does the model transform inputs into outputs?

Suppose the AI model is represented as a function $f : \mathbf{X} \to \mathbf{Y}$. Without causal information, we can estimate the conditional distribution $P(\mathbf{Y} \mid \mathbf{X})$ from observational data. However, this distribution reflects statistical associations, not necessarily causal effects. Potential issues include:

- **Confounding**: A hidden variable $Z \in \mathbf{V}$ (or $Z \in \mathbf{U}$) affects both $\mathbf{X}$ and $\mathbf{Y}$, inducing spurious associations.
- **Reverse Causation**: The true causal direction might be $\mathbf{Y} \to \mathbf{X}$.
- **Feedback Loops**: Cyclic dependencies complicate the interpretation of $P(\mathbf{Y} \mid \mathbf{X})$.

Without the causal graph $\mathcal{G}$, we cannot compute the interventional distribution: $P(\mathbf{Y} \mid \mathrm{do}(\mathbf{X} = \mathbf{x}))$ which reflects the causal effect of setting $\mathbf{X}$ to $\mathbf{x}$.

### Q4: How do the model's internal mechanisms function?

Internal mechanisms involve the causal interactions among hidden or intermediate variables within the model. Let $\mathbf{H} \subseteq \mathbf{V}$ represent internal variables (e.g., hidden layers in a neural network). The structural equations for $\mathbf{H}$ and their causal relationships with $\mathbf{X}$ and $\mathbf{Y}$ are given by: $H_j = f_{H_j}(\mathrm{pa}(H_j), U_{H_j})$ Without knowledge of $\mathcal{M}$, we cannot specify these equations or the causal graph $\mathcal{G}$, preventing us from understanding how $\mathbf{H}$ mediates between $\mathbf{X}$ and $\mathbf{Y}$.

### Q5: Why does the model make a specific decision for a given input?

Explaining a specific decision requires identifying the causal factors that led from the input $\mathbf{X} = \mathbf{x}$ to the output $\mathbf{Y} = \mathbf{y}$. To perform this explanation, we need to:

1. **Abduction**: Infer the exogenous variables $\mathbf{U} = \mathbf{u}$ consistent with $\mathbf{X} = \mathbf{x}$ and $\mathbf{Y} = \mathbf{y}$.
2. **Trace Causal Pathways**: Use the structural equations to identify how changing $\mathbf{X}$ affects $\mathbf{Y}$.

Without $\mathcal{M}$, we cannot perform abduction because $\mathbf{U}$ and $\mathbf{F}$ are unknown. Additionally, we cannot trace causal pathways without the causal graph $\mathcal{G}$.

### Q6: Why would the decision differ if the input had been different?

Answering this question requires **counterfactual reasoning**, which involves considering a hypothetical scenario where the input is $\mathbf{X} = \mathbf{x}'$ (different from the observed $\mathbf{X} = \mathbf{x}$) and determining the corresponding output $\mathbf{Y}_{\mathbf{X}=\mathbf{x}'}(\mathbf{u})$. As per Pearl (2009), computing counterfactuals involves:

1. **Abduction**: Infer $\mathbf{U} = \mathbf{u}$ from the observed data $(\mathbf{X} = \mathbf{x}, \mathbf{Y} = \mathbf{y})$.
2. **Action**: Modify the structural equations to reflect the counterfactual intervention $\mathrm{do}(\mathbf{X} = \mathbf{x}')$.
3. **Prediction**: Compute the counterfactual outcome $\mathbf{Y}_{\mathbf{X}=\mathbf{x}'}(\mathbf{u})$ using the modified model.

Without the SCM $\mathcal{M}$, none of these steps can be performed accurately.

**Overall Conclusion** The absence of causal information—specifically, the structural causal model $\mathcal{M}$—restricts us to the observational distribution $P(V)$, preventing us from identifying underlying data-generating mechanisms, understanding causal pathways within the model, and performing counterfactual reasoning. Consequently, causal information is essential for providing accurate and reliable explanations in XAI. Without it, explanations may be incomplete, incorrect, or misleading. ∎

