# OpenReview forum: "Position: Explainable AI is Causal Discovery in Disguise"
_ICML.cc/2025/Position_Paper_Track — Submitted to ICML 2025 Position Paper Track_

### Official Review · Reviewer_MLVr · 2025-03-09

**Significance:** 3
**Argument Clarity:** 2
**Rating:** 3
**Confidence:** 3

**Questions:**

My questions are derived from the list of weaknesses above:

1. Can you clarify what your position is? Do you only focus on causal discovery (what the title suggests to me) or causality at large (what the content suggests to me)?
2. What is (a clear formulation/example of) the alternative view you are you are arguing against?
3. What would be a concrete recommendation you derive from your position? By concrete, I mean something that a researcher could start investigating (eg, a project idea or a research question).

**Discussion Potential:**

3

**Paper Summary:**

The authors argue that the field of explainable AI is aiming to answer causal (observational, interventional, and counterfactual) questions. Thus, bringing causal modeling into xAI is recommended - in the authors' words, necessary and sufficient - to answer the questions the xAI community asks.

The paper provides
- background on xAi (sec 2)
- background on causality (sec 3)
- reasons about the necessity and sufficiency of causality for answering xAI questions (sec 4)
- proposes an agenda to move the field forward by focusing on concept and relation discovery, discusses the practical limitations of obtaining causal models, and discusses the alternative view (sec 5)

**Position:**

Yes

**Position In Title:**

Yes

**Related Work:**

2

**Strengths And Weaknesses:**

## Strengths
The paper's position (more precisely, the steelmanned version; for explanation see my elaboration below at the weaknesses) is timely, reasonable, and interesting.
Also, **I believe finding underlying connections between different (machine learning) fields can be a valuable contribution. The overview of xAi methods and their categorization (and illustration with Figure 1) is intuitive and easy-to-follow.**

The paper has some very nice takeaways, including:

- "a more nuanced view is that the required level of causal grounding depends on the stakeholder’s objective" - for this, see also [(Marconato et al., 2023)](https://www.mdpi.com/1099-4300/25/12/1574)
- "concepts should be identified at a granularity that is useful to a given stakeholder (or audience). Even if we had a perfect SCM of low-level features (e.g., pixels), explanations would remain unhelpful unless translated to higher-level abstractions that align with human mental model"
- "Concept discovery should align with users’ internal models, recognizing that no single “correct” variable decomposition exists universally (Teso et al., 2023)."

## Weaknesses

The paper's position hits a relevant and timely point, though there are weaknesses involved:
- **As stated currently, the position does not reflect what the authors advocate for**. "Causal discovery" is only about learning the causal relationships (the DAG), whereas the authors also discuss counterfactuals. I suppose what the authors mean is "xAI is Causality in Disguise"
- The position paper misses very relevant recent works advocating for the causality in xAI, e.g. [(Marconato et al., 2023)](https://www.mdpi.com/1099-4300/25/12/1574)
- Starting with the background on xAI already relies on causal concepts, which are only introduced later on, making it potentially harder for readers from the xAI community (assuming they are not familiar with causality)
- The background on causality contains inaccuracies and a few - in my opinion - suboptimal choices
	- The section is a list of definitions (in Def. 3.4, it is hard for me to figure out what it defines), which is probably hard to follow for people unfamiliar with causality
	- The exogenous variables are not necessarily unobserved, and the endogenous ones are not necessarily observed (Def 3.1)
	- Def. 3.3. should be accompanied by a visualization. Explaining causal concepts without a graph is not conducive to understanding. Please add examples
- **Sec. 4 is the most problematic in the paper**:
	- My concern with Def. 4.1 is that it does not seem to be operationalizable. If I were working on xAI, I could not do anything with it (maybe this wasn't the purpose)
	- Again, listing definitions and theorems misses the opportunity to form a convincing narrative, which is probably even more critical for a position paper. I can guess, but I do not know what the authors intended to say with these.
	- Thm. 4.2. assumes a "unique true" SCM, which needs to be discussed. Non-identifiability results posit that without assumptions, this is impossible (example: scaling the structural equations by 2, and dividing the noise variables by 2 gives a causal model, which is in many, but not all aspects equivalent, and most algorithms for learning causal models will not be able to distinguish the two)
	- The law of structural models is not referred to (please cite a source)
	- **The proof Thm 4.3 proves a different statement than Thm 4.3.**
		- The current proof gives an example but does not show that if we do not have access ot the model, we cannot answer Q1-Q6
	- 4.1
		- For the sample complexity and scalability concerns, recent results in score-based causal discovery could be relevant to discuss, such as [(Montagna et al., 2024)](http://arxiv.org/abs/2407.18755) and previous works
		- Identifiability and equivalence classes were not defined before being mentioned. I agree that they are an essential part of the discussion. My concern is that people with no background in identifiability theory will have a hard time understanding the importance/meaning of these concepts
- 5.3
	- I value the structured recommendations by the authors, and I wish they could make it more concrete, as my concerns with the current formulation is that it is hard to turn it into a concrete idea
- 5.4
	- please add citations for causal abstraction models and neuro-symbolic reasoning
	- **It is unclear to me from the current formulation what the alternative view is**: the included high-level example seemingly does not contradict the SCM framework (objects, attributes, forces, can also be nodes in a DAG) - based on the abstract of the cited Gerstenberg and Tenenbaum book chapter, they also seem to argue for causal modeling, and not against it
	- it is unclear what the authors mean by "queries that are difficult to formalize in an SCM, where variables typically represent discrete events or predefined states." As far as I know, discrete variables can also be incorporated into the SCM framework (e.g., switching models in [(Sontakke et al, 2021)](http://arxiv.org/abs/2010.03110) or [identifiability results for Switching Dynamical Systems](http://arxiv.org/abs/2305.15925))

**Support:**

2

---

> ### Author Rebuttal · Authors · 2025-03-30
>
> We sincerely thank Reviewer MLVr for their thorough and insightful comments. We particularly appreciate your recognition that our position, in its "steelmanned version," is "timely, reasonable, and interesting" with "some very nice takeaways." Reviewer cL7i similarly emphasizes the clear benefits of adopting causality as a principled framework, underscoring the strength of our argument.
>
>
> ### Response to Questions:
>
>
> **A1:** Thank you for highlighting this ambiguity. Our primary position is broader than causal discovery alone; it argues explicitly for framing XAI through general causal reasoning, including observational, interventional, and counterfactual queries. Causal discovery specifically refers to identifying causal structures, but our argument encompasses the broader application of causal semantics to unify and clarify XAI objectives. Thus, we agree the title might currently imply a narrower scope than intended. We are open to more descriptive alternative titles, such as:
>
> + "XAI is Causality in Disguise"
> + "Explainable AI Needs a Causal Discovery Framework"
> + "Unifying Explainable AI from a Causal Perspective"
>
>
> We welcome your input on these alternatives.
>
>
> **A2:** An explicit recommendation from our framework involves investigating methods for causal abstraction—discovering intermediate-level concepts and relationships aligned with human mental models and suitable for answering causal queries. A concrete research direction could be: developing and evaluating algorithms that extract meaningful causal abstractions from complex neural network representations (e.g., Geiger et al. as referenced). This includes bridging causal abstraction methods with neuro-symbolic reasoning to ensure abstractions remain both causally accurate and cognitively intuitive.
>
> Another practical recommendation is explicitly studying approximate causal modeling under computational constraints. Future research could focus on auantifying and systematically analyzing trade-offs between causal explanation accuracy and computational complexity, particularly developing \epsilon-approximate causal explanations that retain practical utility.
>
> These research directions would concretely operationalize our position’s theoretical stance into measurable, impactful contributions.
>
>
> ### Discussion about Weaknesses:
>
> **Sequence of Causality and XAI Sections:**
> Your observation regarding the ordering of background sections is valuable. We acknowledge the current structure might initially confuse readers unfamiliar with causality. We will explicitly streamline and possibly reorder these sections to introduce causal concepts earlier and clearly motivate subsequent XAI discussions.
>
> **List of Definitions (Def. 3.4):**
> Thank you for pointing this out. Both you and Reviewer ti3F have noted this concern. We agree that Definition 3.4, in its current form, lacks clarity for readers less familiar with causality. We will revise this definition to clearly define the relevant concept and ensure intuitive accessibility.
>
> **Exogenous and Endogenous Variables (Def. 3.1):**
> You rightly pointed out our oversight. Exogenous variables are not necessarily unobserved, nor are endogenous variables necessarily observed. To clarify this explicitly, we will adopt the clear example suggested by Reviewer ti3F (`SunriseTime` → `MoodToday` ← `SleepHours`) and correct Definition 3.1 accordingly.
>
> **Visualization of Causal Concepts (Def. 3.3):**
> We fully agree that causal concepts benefit significantly from visualization. Following your and Reviewer ti3F’s suggestion, we will explicitly include visual examples—such as the `SunriseTime` → `MoodToday` ← `SleepHours` scenario—either in the main text or as supplementary material, enhancing intuitive understanding for broader readers.
>
> **Operationalizability (Def. 4.1):**
> We appreciate your concern regarding the operationalizability of Definition 4.1. Indeed, operationalizing this stance is challenging, as explicitly noted in §4.1 and throughout §5. Our primary purpose is to frame XAI clearly in causal terms to help researchers understand the clear possibility/impossibility landscape and motivate targeted causal research.
>
> **Clarifying the Alternative View:**
> Your comment regarding the alternative view raised by Gerstenberg and Tenenbaum is insightful. Our point was that SCMs might not always represent the most intuitive or effective modeling choice, particularly for certain cognitive or conceptual scenarios. We will explicitly clarify this alternative perspective, referencing your observation and further elaborating as discussed with Reviewer ti3F.
>
>
> ### Final Remarks:
>
> We respectfully ask Reviewer MLVr if our explicit clarifications and proposed revisions effectively address your concerns. If these revisions adequately respond to your feedback, we would greatly appreciate an increased support rating. We warmly welcome any additional recommendations you may have to further enhance the manuscript.

---

> > ### Comment · Reviewer_MLVr · 2025-04-02
> >
> > I thank for your detailed response! Given the importance and the high discussion potential of the work, I will increase my score to 3.
> >
> > ## A1: title
> > I think the first or third one you proposed would be a better fit.
> >
> > ## A2: research directions (minor)
> > I appreciate your suggestions. I believe this is largely a subjective judgment of mine: they still seem to be not concrete enough. What I would like to see is a list of questions a new graduate student could start working on.
> >
> > ## Discussion and weaknesses
> > I appreciate your explanation

---

> > > ### Author Response · Authors · 2025-04-08
> > >
> > > Thank you again, Reviewer MLVr, for increasing your score and appreciating the discussion potential of our paper.
> > >
> > > We value your request for concrete, actionable research questions suitable for new graduate students. As such, below, we provide some research questions aligned with our position, the discussed literature, and practical concerns highlighted in your feedback. Given the purpose of the ICML position paper track—to "inspire constructive, useful discussion within the ICML community"—we believe adding these questions (embedded in the main body, space permitting, or elaborated in an appendix) will effectively serve this objective. Thank you again for your suggestions.
> > >
> > > ## Concrete Research Questions for PhD Students:
> > >
> > >
> > > ### Causal Abstraction and Representation Learning
> > >
> > > 1. How can we systematically identify and extract human-interpretable concepts from deep neural networks (DNNs) to construct meaningful causal abstractions?
> > > > Approach: Develop unsupervised or semi-supervised algorithms leveraging neuro-symbolic methods to identify concept-level abstractions from latent spaces.
> > >  (Related Work: Geiger et al. (ICLR, 2021); Teso et al. (NeurIPS, 2023); Zhang (2024).)
> > >
> > > 2. Under what conditions can we guarantee that causal abstractions discovered from low-level neural representations remain faithful to the original neural network's causal structure?
> > > > Approach: Study theoretical guarantees for causal abstraction under approximate causal modeling frameworks. Quantify trade-offs between abstraction fidelity and interpretability.
> > >  (Related Work: Beckers & Halpern (AAAI, 2019); Rubenstein et al. (UAI, 2017); Karimi et al. (ICML, 2024).)
> > >
> > >
> > > ### Approximate and Partial Causal Models
> > >
> > > 3. How can we formulate and efficiently compute approximate causal explanations that balance computational complexity with explanation accuracy?
> > > > Approach: Formalize a notion of \epsilon-approximate causal explanations and develop computational algorithms (e.g., scalable score-based discovery methods) to practically derive such explanations from large-scale neural network models.
> > >  (Related Work: Montagna et al. (CLeaR, 2025); Glymour et al. (Frontiers in Genetics, 2019).)
> > >
> > > 4. Can we establish rigorous bounds on how approximations in causal discovery methods affect downstream XAI tasks such as model debugging or user trust?
> > > > Approach: Empirically and theoretically analyze sensitivity to approximation errors for common XAI tasks, developing formal metrics and robustness measures.
> > >  (Related Work: Janzing et al. (AISTATS, 2020).)
> > >
> > >
> > > ### Human-Aligned Causal Queries and Interactive XAI
> > >
> > > 5. How should interactive XAI systems leverage causal interventions to enhance users' mental models about AI systems?
> > > > Approach: Design and experimentally validate interactive interfaces (e.g., conversational agents or visual analytics tools) that enable users to perform causal interventions, systematically studying the impact on user understanding.
> > >  (Related Work: Miller (2019))
> > >
> > > 6. Which classes of causal queries (observational, interventional, counterfactual) are most effective in various stakeholder contexts (e.g., healthcare, finance, autonomous systems)?
> > > > Approach: Conduct domain-specific user studies to measure how different causal query types influence trust, decision quality, and transparency perceptions across diverse stakeholder groups.
> > >  (Related Work: Wachter et al. (Harvard Journal of Law & Tech, 2018); Mittelstadt et al. (FAccT, 2019).)
> > >
> > > We sincerely hope these detailed, immediately actionable research questions fulfill your expectations and demonstrate a clear pathway from our theoretical position towards tangible future contributions to XAI research.
> > >
> > >
> > > ## References
> > >
> > > "Causal Abstraction: A Theoretical Foundation for Mechanistic Interpretability," Geiger et al., 2023
> > >
> > > "Abstracting Causal Models," Beckers & Halpern, AAAI, 2019
> > >
> > > "Causal Consistency of Structural Equation Models," Rubenstein et al., UAI, 2017
> > >
> > > "Score Matching Through the Roof: Linear, Nonlinear, and Latent Variables Causal Discovery," Montagna et al., CLeaR, 2025
> > >
> > > "Review of Causal Discovery Methods Based on Graphical Models," Glymour et al., Frontiers in Genetics, 2019
> > >
> > > "Feature Relevance Quantification in Explainable AI: A Causal Perspective," Janzing et al., AISTATS, 2020
> > >
> > > "Explanation in Artificial Intelligence: Insights from the Social Sciences," Miller, AI Journal, 2019
> > >
> > > "Counterfactual Explanations without Opening the Black Box," Wachter et al., Harvard Journal of Law & Tech, 2018
> > >
> > > "Explaining Explanations in AI," Mittelstadt et al., FAccT, 2019
> > >
> > > "Causal Abstraction in Model Interpretability: A Compact Survey," Zhang, preprint, 2024
> > >
> > > "On the Relationship Between Explanation and Prediction: A Causal View," Karimi et al., ICML, 2024

---

### Official Review · Reviewer_cL7i · 2025-03-13

**Significance:** 3
**Argument Clarity:** 4
**Rating:** 4
**Confidence:** 5

**Questions:**

__Q1.__ Regarding weakness W1,  a relevant question is, how can practitioners deal with these intrinsic obstacles? Can we design more relaxed causal queries that might nevertheless be useful in practice? Consider, for instance, the requirement that the answers to the core XAI questions be accurate and complete. This requirement is quite stringent. It is possible that relaxing these requirements still produces answers that are practically useful even though they are not ideal.

__Q2.__  How does your classification scheme relate to terminology such as 'global' and 'local' queries? (e.g., Barcelo et al., 2020)
For instance, are global input queries "model based" or "data based"? It seems they could be categorized either way.

__Q3.__ Is the notation in Definition 3.1, bullet point 3 meant as follows? $\mathbf{F} = \\{ f_V : V \in \mathbf{V} \\}$.
If so, you might want to use this standard notation instead. If not, perhaps a clarification is needed.

**Discussion Potential:**

3

**Paper Summary:**

This paper proposes that XAI's goals and methods would be better understood and unified by framing them in the language of *causal discovery*.
It provides a conceptual framework that classifies XAI queries into 3 categories: data-based queries, model-based queries, and decision-based queries.
The authors argue that these correspond, respectively, to observational correctness, interventional correctness, and counterfactual correctness in a causality framework.
They also prove the necessity and sufficiency of the causality framework and discuss some limitations and possible directions.

## Update after rebuttal

Thanks for the responses and clarifications regarding concrete changes. I believe these are adequate and strengthen the paper. I’ve raised my confidence score on the accept recommendation.

**Position:**

Yes

**Position In Title:**

Yes

**Related Work:**

3

**Strengths And Weaknesses:**

__Strengths__

__S1.__ The paper is very well written and the position is well argued.

__S2.__ I found it an interesting, informative, constructive and educational read.

__S3.__ The topic and position are relevant to the explainability/interpretability community and they hold potential to clarify and organize efforts in these areas.

__Weaknesses__

__W1.__ The paper proposes an approach to XAI with causal queries that are highly general and idealized. This is _not_ a weakness itself, of course, but such stringent query properties are often in conflict with the intractability of computing answers to the queries.
Note, for instance, that simple explainability/interpretability queries with useful properties are already intractable in arguably less ideal cases than those described by the causal discovery framework (see, e.g.,  Bassan et al., 2024; Adolfi et al., 2025; Barcelo et al., 2020).

__W2.__ The discussion of limitations is good but does not engage much with the literature on the aforementioned issues, in particular, the computational complexity of explainability/interpretability queries (Bassan et al., 2024; Adolfi et al., 2025; Barcelo et al., 2020).

__W3 (minor).__ The position statement — "explainable AI is causal discovery in disguise", which appears in the title and at various places in the main text — suggests that the paper will argue and/or prove an equivalence between what XAI researchers do/want and causal discovery.
I think the benefit of adopting causality as a framework, as well as the necessity and sufficiency of the SCM, is well argued in the paper, but the equivalence between the causality framework and XAI is not.
As I say elsewhere, it seems possible that much lenient queries with more modest properties might still be useful in practice for particular applications.
Moreover, these might not be just possible but in some cases necessary and more plausible than idealized causal queries.
More fundamentally, since XAI as a research domain might be legitimately heterogeneous and can evolve its goals and methods in multiple different directions according to specific domain applications, it is not clear that XAI *is* causal discovery,
If this reasoning is correct, the authors' position might be better characterized as some variant of "explainable AI needs a causal discovery framework" rather than variants of "XAI *is* causal discovery".

__W4 (minor).__ The main argument could be summarized closer to the beginning of the paper; to this reviewer, there's currently a lot of delaying of substantive content.

__W5 (minor).__ The how-why-what classification does not seem like a useful mnemonic or organizational scheme. Since the assignment of the question word (e.g., "what") is arbitrary, one could phrase the "how"-question Q3 as "*what* are the necessary and sufficient...?". Therefore the mapping doesn't seem particularly meaningful.

__W6 (minor).__ p. 3, line 135 mentions causal discovery and its aims but does not define it.

__W7 (minor).__ line 159 could use a citation after "required".

__W8 (minor).__ The reader could be directed to Figure 1 earlier in the main text.

__W9 (minor).__  p. 4, last paragraph of section 2.3 states conclusions without explanation: "...lacking causal assumptions limits robustness and generalizability across contexts"; "...explanation is causal discovery in disguise".

__W10 (minor).__  p. 5, "fidelity" is used before defining it.

__W11 (minor).__  p. 5, Missing definition: "abduction-action-prediction"

__W12 (typo).__  p. 5, "to to"

__W13 (citation needed?).__ p. 7, "granularity" (see Vilas et al., 2024)

__W14.__  You write "In practice, obtaining a fully accurate causal model is often infeasible due to data and computational limitations. To address this, we advocate for approximate causal models". But note that approximate answers often suffer from the same computational infeasibility as exact answers (e.g., see References). Grappling with these potential barriers and engaging with the relevant literature seems warranted.

__W15 (citation needed).__ p. 8, "critics argue that a truly human-aligned XAI framework may require fundamentally different tools—potentially drawing from cognitive science, probabilistic programs, or physics-inspired models—to bridge the gap between mechanistic causality and intuitive human understanding."

__References__

Bassan, S., Amir, G., & Katz, G. (2024). Local vs. Global Interpretability: A Computational Complexity Perspective. Proceedings of the 41st International Conference on Machine Learning, 3133–3167.

Adolfi, F., Vilas, M. G., & Wareham, T. (2025). The Computational Complexity of Circuit Discovery for Inner Interpretability. The Thirteenth International Conference on Learning Representations.

Barceló, P., Monet, M., Pérez, J., & Subercaseaux, B. (2020). Model Interpretability through the lens of Computational Complexity. Advances in Neural Information Processing Systems, 33, 15487–15498.

Vilas, M. G., Adolfi, F., Poeppel, D., & Roig, G. (2024). Position: An Inner Interpretability Framework for AI Inspired by Lessons from Cognitive Neuroscience. Proceedings of the 41st International Conference on Machine Learning, 49506–49522.

**Support:**

4

---

> ### Author Rebuttal · Authors · 2025-03-30
>
> We sincerely thank Reviewer cL7i for their detailed and constructive feedback. We particularly appreciate your recognition of the paper as "interesting, informative, constructive and educational," and your strong endorsement of our causal framework as "well argued." We warmly welcome further discussion.
>
>
> ### Response to Questions:
>
> **A1:** We fully agree with your insightful comment that the causal queries proposed—when instantiated over complete SCMs—can be computationally and epistemically demanding. Indeed, our paper explicitly acknowledges this challenge in §4.1 (“Discussion on Robustness and Limitations”) and §5 (“A Way Forward”). Our stance is not to insist that all XAI methods must provide perfect causal answers. Rather, we argue that causality offers a principled framework that effectively unifies and clarifies XAI’s otherwise fragmented objectives.
>
> To practically handle these inherent obstacles, our paper proposes two explicit relaxation strategies:
>
> + _Approximate Causal Models:_
> We advocate for estimated or partially specified causal models, where not all variables or relationships are fully known. Such approximate causal structures can still offer meaningful improvements over purely correlation-based methods by explicitly aligning explanations with known or hypothesized mechanisms (discussed explicitly in §5.2). Here, accuracy and completeness serve as aspirational goals rather than strict requirements; thus, explanations remain practically useful even when partially grounded.
>
> + _Relaxed Query Semantics:_
> We agree completely with your suggestion about relaxing the stringent accuracy and completeness criteria. For instance, even partial answers, approximate attributions, or probabilistic bounds can yield valuable insights. A counterfactual explanation does not necessarily need perfect accuracy to be practically useful for user trust, debugging, or providing actionable insights (e.g., heuristic recourse or local sensitivity analyses). We believe future research should explicitly explore formalizing a spectrum of causal utility under various tractability constraints, analogous to ε-approximation methods in optimization theory.
>
> Ultimately, our goal is to clarify the aspirational structure of XAI queries using causal semantics, explicitly promoting interactive, approximate, and stakeholder-aligned approaches, which enhance practical accessibility.
>
> **A2:** Thank you for highlighting this useful connection. Indeed, the global/local distinction closely resembles aspects of our taxonomy but does not map perfectly one-to-one. Local interpretability typically refers to explanations about model decisions or internal activations for a single data-point. In contrast, global interpretability usually involves aggregate explanations (e.g., average outputs or activations) across a larger data (sub)population. While global interpretability queries can sometimes seem ambiguous, they typically are categorized as model-based due to their explicit dependence on the model's aggregate behaviors rather than raw data structures alone. We will explicitly clarify this distinction and its implications in §2 of our revision.
>
> **A3:** Indeed, your observation is correct, and we appreciate this detailed feedback. The parentheses were inadvertently omitted. We will promptly correct this notation to the standard form explicitly as you suggested: .
>
>
> ### Additional Clarifications:
>
> **Engagement with computational complexity literature:**
> We appreciate your suggestion to engage further with relevant literature on computational complexity (Bassan et al., 2024; Adolfi et al., 2025; Barceló et al., 2020). We will explicitly incorporate these references into our "Robustness and Limitations" (§4.1) to highlight potential computational barriers more clearly.
>
>
> **Definition clarifications:**
> Your minor suggestions (W6-W15) on missing definitions, citations, and clarifications are gratefully acknowledged and will be explicitly addressed in our revisions. For instance, the missing definition of "abduction-action-prediction" and the premature use of "fidelity" will be corrected and clarified accordingly.
>
>
> **Practical utility and computational limitations:**
> Regarding your insightful concern (W14) about approximate causal answers sharing computational limitations, we fully agree. This point will be explicitly incorporated in §4.1, emphasizing the need for future research to develop methods that effectively balance computational feasibility and explanatory usefulness.
>
>
> ### Final Remarks:
>
> We respectfully ask Reviewer cL7i if these comprehensive clarifications address your concerns sufficiently. If so, we would greatly appreciate an increased support rating. We warmly welcome any additional recommendations or suggestions to further enhance the quality and clarity of our manuscript.

---

> > ### Comment · Reviewer_cL7i · 2025-04-02
> >
> > Thanks for your answers. I believe the proposed changes are adequate. Note that the changes to address some of the main weaknesses (W1, W2, and W14) are only promised and not implemented in the manuscript or exemplified in the response. Therefore, I have no material on which to base an update on my score.

---

> > > ### Author Response · Authors · 2025-04-08
> > >
> > > We sincerely thank Reviewer cL7i for reviewing our response and providing an additional rebuttal comment.
> > >
> > > We politely remind of this year's updated ICML rebuttal process guideline stating:
> > >
> > > > "**No submission updates**: Similar to previous years, the original submission (PDF and supplemental material) cannot be revised in OpenReview during the discussion period"
> > >
> > > Moreover, due to the 5000 character limit in our initial response, we were unable to exemplify explicitly some proposed changes, which we gladly elaborate upon here.
> > >
> > >
> > >
> > > ## RE: W2 (Computational Complexity Perspective)
> > >
> > > We appreciate the reviewer's provided references on the computational complexity perspective in explainable AI—particularly those we were previously unaware of (Bassan et al., 2024; Adolfi et al., 2025; Barceló et al., 2020). We look forward to engaging deeply with this literature in future research. We concur—and hope the reviewer agrees—that exploring intersections between causal and computational complexity perspectives in XAI represents an exciting avenue for future research. While explicitly integrating computational complexity into our causal framing is beyond the scope of this paper (and indeed, the cited references do not yet explicitly leverage a formal causal framework), we believe our position paper may encourage researchers from both communities to explore these intersections.
> > >
> > > In this spirit, we respectfully direct the reviewer to our explicit "Reply Rebuttal Comment" to `Reviewer MLVr`, where we proposed concrete research questions suitable for a new graduate student. Here, we provide additional research questions explicitly bridging the computational complexity and causal frameworks:
> > >
> > > 1. Can we identify computational boundaries (complexity classes) for causal interpretability queries?
> > > > Explore whether observational, interventional, and counterfactual queries within SCM-based XAI frameworks can be classified according to computational complexity classes defined in existing interpretability literature (Bassan et al., 2024; Adolfi et al., 2025). This could inform practical guidelines about which causal queries remain tractable.
> > >
> > > 2. Under what conditions do computationally efficient approximations of causal queries retain sufficient explanatory power?
> > > > Empirically and theoretically analyze how relaxing causal exactness requirements (e.g., \epsilon-approximate causal explanations, inspired by approximation schemes in computational complexity literature) affects computational complexity and the practical utility of XAI methods. (Barceló et al., 2020)
> > >
> > >
> > >
> > > ## Elaboration on Response to W1/Q1/A1
> > >
> > > Due to character limitations, we previously summarized our strategies for dealing with the intrinsic obstacles posed by stringent causal queries. We respectfully refer the reviewer to our "Rebuttal by Authors" and detailed "Reply Rebuttal Comment" to `Reviewer MLVr`, which we summarize briefly here:
> > >
> > > * **Causal Abstraction Methods:** Developing algorithms that extract meaningful causal abstractions from neural network representations to align explanations with human mental models (Geiger et al., 2021).
> > >
> > > * **Approximate Causal Modeling:** Investigating methods for explicitly quantifying trade-offs between explanation accuracy and computational complexity, developing methods to deliver ε-approximate causal explanations while retaining practical interpretability and usability.
> > >
> > > These concrete directions operationalize our theoretical stance into measurable, impactful contributions.
> > >
> > >
> > >
> > > ## Elaboration on Response to W14 (Approximate Causal Models)
> > >
> > > Regarding our mention of "approximate causal models," we clarify explicitly here that by "approximate," we primarily referred to models that do not exactly match the true causal structure due to epistemic or data limitations—not merely approximations introduced explicitly to reduce computational complexity. We fully acknowledge (as the reviewer insightfully points out) that approximate causal models might also face computational infeasibility. Our recommendation of approximate causal models is intended as a practical recognition that the "true" SCM may rarely be fully accessible or identifiable.
> > >
> > > However, this naturally opens another compelling line of research:
> > >
> > > 3. What computational trade-offs arise when using approximate causal models in realistic XAI scenarios?
> > > > Future research could systematically study whether relaxing accuracy requirements in causal model identification indeed yields tangible computational benefits or if computational complexity remains a core limitation.
> > >
> > > We hope these explicit clarifications and concrete additions directly address the reviewer's concern regarding our previous promises of manuscript improvements. We believe these additions will help meet the stated goal of the ICML position paper track to "inspire constructive, useful discussion within the ICML community," and we are committed to integrating these elements explicitly into our camera-ready version.

---

### Official Review · Reviewer_ti3F · 2025-03-14

**Significance:** 2
**Argument Clarity:** 3
**Rating:** 2
**Confidence:** 4

**Questions:**

Minor question: how does the SCM structure handle exogenous but observed variables (e.g. for me, the time that the sun rises in the morning)?

**Discussion Potential:**

2

**Paper Summary:**

The paper argues that the current XAI literature is "overwhelming", even generating "surveys of surveys", partly as it lacks a clear goal.

The paper claims that using causal discovery as its goal would focus it, presenting necessary and sufficient conditions for a structural causal model (SCM, à la Pearl) to provide answers to the questions asked in XAI:

> Causal assumptions ... are essential to bring coherence to XAI by addressing core questions through a principled lens.

## update after rebuttal

Original scores stand: ICML restrictions on updating manuscripts means I cannot assess how proposed revisions are actually implemented, thus how well they address the concerns raised below.

**Position:**

Yes

**Position In Title:**

Yes

**Related Work:**

3

**Strengths And Weaknesses:**

**STRENGTHS**

I fully agree with the paper's starting point: there is an XAI industry, incentivised by the possibility of publishing novel techniques on niche questions in good ML conferences (thus, incentivised in a slightly different direction than one might think XAI should be directed, namely towards better understanding AI/ML models' predictions in practice).  Hence, this literature can be "overwhelming".

**WEAKNESSES**

The paper may be decomposed into two arguments.  The first, I'll term 'informal', which centers on lists of methods and approaches (e.g. p.3).  The second, 'formal' arguments are the theorems establishing the relationship between SCM and XAI explanations.

My main feeling from the first sort of argument is that it replicated the "overwhelming" concern motivating the paper in the first place.  For example, a dozen methods are presented in the list on p.3 alone.  If we wish to help overwhelmed readers, we need to replace more with fewer, whether by showing that existing methods represent special cases of a more general class, or can be replaced by a smaller, but better founded set of methods.  I did not see this happening.

The theorems, on the other hand, felt somewhat tautological.  If one's ground truth model is a Pearl-style SCM, then any XAI effort to 'explain' must, perforce, explain/discover its causal structure.

Additionally, my superficial impression is that acyclic graphs can only encode a limited subset of causal relations.  Consider, for example, a standard economic problem: consumers choose quantities of items based on prices; firms choose prices based on consumer demand.  In such systems, quantities and prices are determined simultaneously.  Similarly, in a rational expectations equilibrium, agents form expectations about future variables, and then take actions that, collectively, cause their expectations to be, on average, correct.

I do not know whether such standard interactions can be encoded with SCMs.  I do know both that concerns with Pearl's SCM have been expressed by eminent champions of potential outcomes - e.g. Imbens' "potential outcomes and DAG approaches to causality" - but also that are results relating the two approaches (e.g. Ibeling & Icard, 2023).

(Relatedly, §3 opens with the remark that the various frameworks for modeling causality offer "unique perspectives".  This feels lazy, and does not inform the reader: specifically, what can PO do/not do that SCM can/cannot?)

To convince me, as a reasonably knowledgeable reader, I need to see these sorts of concerns addressed.  The paper does not consider them.  (Imbens' comparison also raised concerns about whether Pearl's approach could scale: has that concern been satisfactorily resolved?)

Further, closing off my comments on the theorems, as there is a way of seeing them as the main 'hard' contribution of the paper, they should be introduced properly, and intuitions provided: I was surprised to discover them.

It would also have been nice to see more attention paid to existing work in causal XAI - q.v. Heskes et al., and Frye et al., both at NeurIPS 2020).  Are these useful?  Can they be built upon?  If so, how?  If not, how do these concrete efforts to incorporate causal structures into XAI need to be adjusted?

The paper divides its questions into data-, model- and decision-based questions.  At time, these distinctions seem _ad hoc_: for example, in the data-based section §2.1, it is claimed that revealing the data structure can uncover "influential features".  I do not know what this means in the absence of a model: merely observing Tyler Vigen's famous correlations does not tell us which features influence which.

§2.2 claims that "model-based interpretability is essential in regulatory and high-stakes environments".  A proper argument would address the Lecun/Weinberger position in the 2017 NeurIPS interpretability debate: no, regulators don't need to know _how_ something works; they just need to know that it detects cancer more accurately and safely; once a few years of data come in showing this, they'll approve it.

Similarly, "transparent explanations" in practice turn out to require relatively little: as I understand it, even stringent EU regulation is satisfied by transparent explanation that AI/ML models were used in making a decision, with options for recourse in the event of adverse high-stakes decisions.  Again, to convince an aware, but not expert reader, issues like this need to be engaged with.

Minor pet peeve: when enumerating items (e.g. "three primary types of queries"), use `enumerate` not `itemize`.

Definition 3.4 feels more like a note or comment than a definition.

In §5.1, I can understand how TCAV "cannot discover new, relevant concepts ... unknown unknowns".  I would have found it useful to see how this contrasts to other techniques that can discover unknown unknowns (unsupervised techniques?).

§5.3 presents a research agenda.  Such calls have been historically useful (e.g. in the extreme, Hilbert's questions).  The call, I think, would be more effective if it posed tightly defined open questions: calling for "methods to better handle high-dimensional data" provides no finish line after which a runner can claim success.

**Support:**

2

---

> ### Author Rebuttal · Authors · 2025-03-30
>
> We sincerely thank Reviewer ti3F for their detailed and insightful feedback. We particularly appreciate your agreement with our primary motivation, highlighting the issue of the "overwhelming" nature of current XAI literature. We warmly welcome further discussion.
>
>
> ### Response to Questions:
>
> **A1:** In SCMs, an observed exogenous variable like SunriseTime is treated explicitly as a root node without a structural equation, and its deterministic observed value directly feeds into the structural equations of its descendants. For example, consider the causal structure: `SunriseTime` → `MoodToday` ← `SleepHours`. Here, `MoodToday` can be modeled as `MoodToday` = f(`SunriseTime`, `SleepHours`, `U`), where `SunriseTime` is observed and fixed, `SleepHours` may or may not be observed, and U is unobserved noise.
>
>
> ### Discussion about Weaknesses:
>
>
> **Complexity and Categorization:**
>
> We appreciate your concern regarding the potentially overwhelming list of methods provided (e.g., on page 3). We clarify that our intention was explicitly not to overwhelm the reader, but rather to provide a simple categorization/taxonomy—distinct from yet another survey—that aligns neatly with causal (observational, interventional, and counterfactual) queries. We fully agree that illustrating existing methods as special cases of more general classes would be valuable, although it was not the main focus of this paper. As an example, in §2.2, we note briefly that methods such as LIME can indeed be approximated via input gradients in sufficiently smooth regions. We will clearly state this to help simplify our taxonomy.
>
>
> **Tautological Nature of Theorems:**
>
> We recognize your perception of our theorems as somewhat tautological. Our main intent may not have come across strongly enough in the submission. Our key message, as clarified in our overall response, is not to argue that explanations must always be articulated strictly at the causal level. Rather, we propose that as a research community, focusing on identifying the causal structure will significantly clarify and unify efforts in addressing XAI queries. We welcome any feedback you may have to make this message more clear.
>
>
> **Limitations of DAGs and Cyclic Relations:**
>
> We fully agree with your comment about Directed Acyclic Graphs (DAGs) limiting the full modeling capacity of real-world processes—particularly in contexts with simultaneous and cyclic causal interactions (e.g., economic equilibria). In fact, SCMs are fully capable of handling cycles; DAGs simply constitute a computationally convenient subset that readily facilitates inference. We appreciate your suggestion and will clarify this explicitly in the paper text, particularly in Definition 3.2.
>
>
> **Comparison between Potential Outcomes (PO) and SCM Frameworks:**
>
> Thank you for this comment. While the focus of our position paper naturally aligns with the SCM framework—given its direct compatibility with causal discovery—we recognize the importance of acknowledging the PO perspective explicitly. Therefore, we will incorporate references to provide readers with a balanced perspective:
>
> * Imbens, G. W. (2020). "Potential Outcome and Directed Acyclic Graph Approaches to Causality: Relevance for Empirical Practice." https://doi.org/10.1257/jel.20191547
>
> * Pearl, J. (2018). "The Seven Tools of Causal Inference, with Reflections on Machine Learning." https://doi.org/10.1145/3241036
>
> Additionally, we will explicitly include Imbens' concerns in the alternative views section (§5.4), highlighting that SCM may not always represent the ideal framing.
>
>
> **Engagement with Existing Causal XAI Work:**
>
> Thank you for highlighting additional relevant literature, such as Heskes et al. and Frye et al. (NeurIPS 2020). We indeed considered these, alongside others (e.g., Janzing et al., AISTATS 2020), but faced challenges integrating them without disrupting the generality of our overarching causal framework. Nonetheless, we welcome and appreciate specific suggestions from reviewers on precisely how these additional works could enhance our manuscript’s comprehensiveness and readability.
>
>
> **Regulatory Context Clarification:*
>
> We greatly appreciate your comment regarding regulatory interpretability requirements and the Lecun/Weinberger position from the NeurIPS 2017 interpretability debate. Indeed, regulators often prioritize empirical evidence of performance and safety over detailed internal mechanisms. Our paper aims primarily at guiding future research toward principled frameworks rather than immediate regulatory compliance. Nevertheless, we agree your point is worth explicitly mentioning and will incorporate it into the takeaway section of §4.
>
>
> ### Final Remarks:
> We respectfully ask Reviewer ti3F if these clarifications have effectively addressed your concerns. If so, we would greatly appreciate an improved support rating. Additionally, we warmly welcome any further suggestions for enhancing the clarity and strength of our manuscript.

---

> > ### Comment · Reviewer_ti3F · 2025-04-09
> >
> > Thank you for your replies.
> >
> > Using your notation:
> >
> > **A1**: noted with thanks.
> >
> > **complexity**: I would need to see what the revised manuscript looks like.  The basic difficulty with the current approach is that it raises concerns about 'overwhelming' readers - and then proceeds to overwhelm readers.
> >
> > **theorems**: theorems prove results; surrounding text motivate and explain those results.  Thus, theorems about DAGs are just that.  If your overarching message is that XAI makes more sense in the light of causal knowledge, it's not clear to me that this is a consequence of your theorems: (1) causal structure is much more general than that which acyclic graphs can encode; (2) the theorems themselves then need to be applied to problems to _show_  that they shed light on applications.  Seen this way, the theorems seem distracting to your message.
> >
> > **limitations of DAGs**: as above; if your theorems are restricted to acyclic causal structures, and are used to provide a grounding for the surrounding text, then the surrounding text/message is also limited to acyclic structures.  I have not re-read the manuscript to see if it clearly restricts itself this way, but don't recall that it does.  If it did, it would need to discuss the limitation.
> >
> > **PO & SCM**: again, I would need to see the revised text.  My concern would be that the Imbens-Pearl debate raises substantial issues; a position paper should walk readers through the relevant issues, to help them reach a conclusion.  This therefore requires more than just a few references.
> >
> > **existing causal XAI**: there are famous comments about the relative roles of proofs and testing in computer science (e.g. Knuth's, "Beware of bugs in the above code; I have only proved it correct, not tried it").  Thus, if you want to convince readers that causal knowledge clarifies and unifies XAI explanations, then I think you need to show them examples of this.  What tools do we have?  What do they help us understand?
> >
> > **regulatory environment**: noted.

---

> > > ### Author Response · Authors · 2025-04-09
> > >
> > > We sincerely thank you for continuing the discussion and taking the time to engage with our rebuttal. We recognize and appreciate the thoughtful follow-up you provided, especially given the many demands on reviewers’ time. Since the response window closes very shortly, we wanted to offer a brief clarification and outline possible next steps for improving the clarity and focus of the paper.
> > >
> > > We also respectfully note this year's updated ICML rebuttal process guidelines:
> > >
> > > > "No submission updates: Similar to previous years, the original submission (PDF and supplemental material) cannot be revised in OpenReview during the discussion period."
> > >
> > > This constraint, along with the 5000-character limit in the initial response, limited our ability to incorporate and exemplify proposed changes, which we would otherwise welcome the opportunity to explore further.
> > >
> > >
> > > ### On Overwhelming Content and the Paper’s Framing
> > >
> > > We appreciate your concern that the manuscript might still feel overwhelming, despite its stated intention to address that very issue. As you note, the paper opens by identifying the fragmentation and excess of the current XAI literature. Our aim is to offer a unifying lens through the causal framework (kindly see our position in bold on pp.1-2)—not to produce a comprehensive survey, but to realign the community’s focus. The examples on p.3 are meant to be illustrative, not exhaustive.
> > >
> > > Still, we understand your concern and would welcome your thoughts on these two possible refinements:
> > >
> > > 1. **Reducing the density of methods in §2**, or moving detailed lists to the appendix, to ensure readers aren’t distracted from the core conceptual position.
> > >
> > > 2. **Changing the paper’s title** to more clearly reflect its unifying, rather than comprehensive, intent. For example, titles like _“Unifying Explainable AI from a Causal Perspective”_ or _“Explainable AI Needs a Causal Framework”_ (suggested in our response to `Reviewer MLVr`) may more accurately signal our goal.
> > >
> > > We would be grateful for your feedback on either direction.
> > >
> > >
> > > ### On DAGs, Theorems, and Framework Scope
> > >
> > > We appreciate your insight that the formal theorems (restricted to DAG-based SCMs) might constrain the broader conceptual message. Our goal was to clarify how, within the assumptions of the SCM framework, XAI queries correspond to causal queries. We agree that future work should extend this to cyclic or dynamic systems, and we will more clearly state the scope of our theorems and the limitations imposed by those assumptions.
> > >
> > >
> > > ### On PO vs. SCM and the Need for Depth
> > >
> > > We agree that the Imbens-Pearl debate deserves a more substantive walkthrough than currently provided. Given space constraints, we had hoped to gesture at the broader conversation, but we now recognize that a position paper should help readers understand competing frameworks. If accepted, we would welcome the chance to include a focused comparison that walks readers through core contrasts.
> > >
> > >
> > > ### On Concrete Tools and Examples
> > >
> > > Your call to provide more concrete tools or demonstrations of causal clarity in practice is well taken. In the spirit of Knuth’s remark you quoted, we are keen to integrate concrete causal XAI methods (e.g., Frye et al., Heskes et al.) as worked examples showing how causal structure can unify explanation approaches. We agree that this will ground our position and welcome the chance to add these in the revision.
> > >
> > > In the end, we hope the paper’s central goal—to inspire constructive and unifying discussion within the ICML community—is evident and aligned with the purpose of the position paper track. We thank you again for your thoughtful comments and sincerely hope this dialogue continues.

---

### Official Review · Reviewer_kJSH · 2025-03-14

**Significance:** 2
**Argument Clarity:** 2
**Rating:** 3
**Confidence:** 4

**Questions:**

Can the authors provide a simple definition of causality up-front to help interpret all the claims that come later? It is not always clear to me what methods the authors consider to be causal or not. Dimensionality reduction is even described as 'aligning closely with the principles of causal discovery'. So what methods are *not* causal? If the claim of the paper is that taking a causal approach will help us, then causality has to have a definition that actually has teeth, and excludes certain lines of work.

Relatedly, the authors seem to acknowledge that there are many cases where non-causal methods are appropriate for XAI problems. So in what way does this causal framework unify the whole field? (e.g. "In light of these challenges,correlation-based explanations (e.g., feature importances, saliency maps) may suffice when the goal is to detect patterns, biases, or anomalies rather than to enable interventions.")

I am confused by the 6 question framework. Q1 and Q2 claim to be about the data generating process and yet the methods discussed (attention and DR) are about internal model mechanisms and a way of *describing* data structure, but not explaining how it is generated. Furthermore, I don't think XAI is actually used to try to explain how data is *generated*. It wouldn't make sense to analyze models in isolation to try to understand the processes in the real world that created the data (and it is simply a different problem than what XAI focuses on).

Furthermore Q4 asks about the how the model's internal functions work yet the methods listed don't analyze the internal mechanisms of the models. The attention analysis listed under Qs 1&2 is actually more appropriate as a means of explain internal model mechanisms (as are many other methods from mechanistic interpretability).

The authors talk about the difficulty of causal discovery. In the case of understanding a trained network, however, we have direct access to the causal model - it is the trained network itself. So it doesn't seem like causal discovery itself is the challenge (and in that way, XAI is not just causal discovery in disguise). Rather, the challenge of XAI is how to create the right abstractions and approximations that are useful for a person or use-case, but don't deviate too much so as not to provide accurate intuitions. The authors talk about this need to find the right concepts that 'align with human mental models', but it is not at clear how this would be evaluated within the causal framework. If our understanding of a model is defined in terms of abstract approximations to what the model is actually doing, how do we evaluate our understanding based on observational, interventional, and counterfactual data? Essentially, we will create a causal graph where the Fs have no real counterpart in the real model.

In total, the paper acknowledges many of the challenges of XAI, but doesn't make clear how causality is uniquely well-suited to help us tackle them. In fact, I think there are many sentences in the paper where removing the word 'causal' would hardly change the meaning. I would like to have a clearer understanding of taking this framing helps, beyond just a label.

**Discussion Potential:**

2

**Paper Summary:**

This paper offers a framework for organizing questions in XAI and claims that thinking of XAI in terms of causality can help unify the field. A formal description of causal models and their operations is given, as is the relationship to the XAI questions defined. Potentially challenges, especially associated with causal discovery are discussed.

**Position:**

Yes

**Position In Title:**

Yes

**Related Work:**

4

**Strengths And Weaknesses:**

Strengths:

-Writing is clear

-Tackles relevant problem (the disorder of XAI)

-Cites relevant work

Weaknesses:

-The organization of the question framework is confusing (see Questions below)

-While the paper addresses many challenges in XAI, it is still not clear to me how the causal framework unifies these things or even how it is a uniquely well-suited framework for tackling these challenges. (see Questions below)

-The framing of XAI as being about discovering the causal model that governs *the world* seems misguided. XAI is about understanding model behavior.

**Support:**

3

---

> ### Author Rebuttal · Authors · 2025-03-30
>
> We sincerely thank Reviewer kJSH for their thoughtful feedback. We particularly appreciate your recognition of our paper’s clear writing, relevance, and strong engagement with related work.
>
>
> ### Discussion About Weaknesses:
>
> **Causal Discovery vs. Abstraction Level:**
> We fully agree with your observation regarding causal discovery within trained networks. Indeed, the fundamental challenge in XAI is not merely causal discovery per se but rather discovering the appropriate level of abstraction (explicitly discussed in §4.1 and extensively in §5). Aligning discovered concepts with human mental models is precisely the critical challenge we emphasize. We will explicitly reiterate and highlight this point clearly in our revision to avoid ambiguity.
>
>
> **Evaluation of Abstract Causal Representations:**
> We appreciate your concern regarding the evaluation of abstract causal representations. Indeed, our paper does not claim to have fully solved this critical issue, given that this is a position paper rather than a comprehensive empirical study. Our primary argument is explicitly that if we have access to a suitable causal representation (be it data-driven, model-driven, or decision-driven), this would directly enable addressing the queries targeted by XAI methods. Our intent is explicitly to encourage the broader research community to pursue concrete strategies for identifying and evaluating these causal representations at appropriate levels of abstraction.
>
>
> **Clarification of the Six-Question Framework:**
> We respectfully disagree with your categorization of questions Q1 and Q2. The methods discussed (e.g., dimensionality reduction and attention mechanisms) are indeed relevant to data-structure analysis independent of a trained model. Specifically, dimensionality reduction techniques explicitly reveal inherent structure within the data itself, independent of any modeling effort. Attention mechanisms, while prominent in transformer-based models, fundamentally serve as general methods for efficiently learning pairwise relationships within sequential or structured data, independent of any trained network. However, we acknowledge that attention analysis can indeed provide insights into model internals when applied post-training, which might have caused confusion. We will explicitly clarify these distinctions in our revision.
>
>
> **Suitability of Causality for XAI:**
> We acknowledge your general concern about explicitly demonstrating the unique suitability of causality for addressing XAI challenges. Our central argument is explicitly aspirational—if we indeed possessed accurate causal models, we could directly and rigorously answer all critical XAI queries, spanning observational, interventional, and counterfactual scenarios. Thus, our paper explicitly motivates the XAI community to focus more systematically on causal methods as a principled foundation. We will explicitly emphasize this aspirational framing clearly in the revision to better articulate our position. As in the response to reviwer MLVr, we are open to, and welcome suggestions for, modifying the paper title if the general agreement is that it would better reflect the position of our work.
>
>
> ### Final Remarks:
>
> We respectfully ask Reviewer kJSH if these explicit clarifications and proposed revisions sufficiently address your concerns. Should these clarifications adequately resolve your main issues, we kindly ask you to consider increasing your support rating. We warmly welcome additional recommendations or suggestions to further enhance our manuscript.

---

> > ### Comment · Reviewer_kJSH · 2025-04-03
> >
> > I thank the authors for their response. Unfortunately this has not clarified much for me, and perhaps introduced more questions (such as how attention mechanisms exist independent of any trained network). I will leave my score as is.

---

### Decision · Program_Chairs · 2025-04-30

**Decision:**

Reject (with encouragement)

**Comment:**

Your paper was highly rated, and we would have liked to accept it this year.  However, due to constraints on conference capacity, we had to reject some papers despite very positive reviews.  The area chair's original meta-review (copied below) is a testament to the strengths of your paper.

----
The reviewer recommendations (one "accept", two "weak accept", and one "weak reject") put this paper in a borderline position, although leaning towards acceptance. After reading the paper, and all the reviews and rebuttal, I think the authors have made a good job at addressing most of the concerns raised by the reviewers, thus I am recommending acceptance.